# A computational offloading optimization scheme based on deep reinforcement learning in perceptual network

Yongli Xing[1], Tao Ye[2], Sami Ullah[3], Muhammad Waqas[4]*, Hisham Alasmary[5], Zihui Liu[1]

**1** School of Sciences, China University of Geosciences, Beijing, China, **2** Faculty of Information Technology, Beijing University of Technology, Beijing, China, **3** Department of Computer Science, Shaheed Benazir Bhutto University, Sheringal, Dir, Pakistan, **4** Department of Computer Engineering, College of Information Technology, University of Bahrain, Al Janabiyah, Bahrain, and also with the School of Engineering, Edith Cowan University, Perth, WA, Australia, **5** Department of Computer Science, College of Computer Science, King Khalid University, Abha, Kingdom of Saudi Arabia

* engr.waqas2079@gmail.com

## Abstract

Currently, the deep integration of the Internet of Things (IoT) and edge computing has improved the computing capability of the IoT perception layer. Existing offloading techniques for edge computing suffer from the single problem of solidifying offloading policies. Based on this, combined with the characteristics of deep reinforcement learning, this paper investigates a computation offloading optimization scheme for the perception layer. The algorithm can adaptively adjust the computational task offloading policy of IoT terminals according to the network changes in the perception layer. Experiments show that the algorithm effectively improves the operational efficiency of the IoT perceptual layer and reduces the average task delay compared with other offloading algorithms.

## 1. Introduction

The Internet of Things (IoT) has very significant advantages over traditional communication technologies. However, IoT devices have limited resources [1, 2], Therefore, we need an alternative unit to perform tasks from end devices and return results in many IoT applications and devices. Generally speaking, limited resources are solved by shifting the computational workload to other devices with better resources and offloading computation [3–5]. With the deep integration of IoTs and edge computing. The original networking mode of IoTs, which focuses on optimizing transmission and saving energy, is increasingly difficult to apply to IoT development. On the other hand, the extensive distribution of edge computing nodes and the powerful offloading capability [6–9] greatly facilitate the task computing and data transmission of IoTs terminals. Therefore, it is essential to study edge computing offloading methods suitable for IoTs.

In IoT edge computing, since the task generation process is highly dynamic, statistics are difficult to obtain or accurately predict [10, 11]. Chen et al. [12] proposed a dynamic computation offloading algorithm based on stochastic optimization. It decomposes the optimization

---

**Data Availability Statement:** The data is available with the link https://github.com/Xing-upup/Unloading-optimization.

**Funding:** The authors extend their appreciation to the Deanship of Scientific Research at King Khalid University for funding this work through large groups project under grant number RGP.2/201/43. The funder provides the funding of this paper as well as the preparation of this paper. None of the authors receive any salary from the funder.

**Competing interests:** The authors have declared that no competing interests exist.

problem into a series of sub-problems and realizes the trade-off between offloading cost and performance. In addition, Zhang et al. [13] proposed a heterogeneous multi-layer mobile edge computing. To support low-latency services, they built a reinforcement learning-based framework to adapt to unstable wireless environments and each edge device's dynamically changing data generation speed. To study a multi-user mobile edge computing network, Li et al. [14] designed an offload strategy based on Deep Q Network (DQN), where users can dynamically adjust the offload ratio to reduce system delay. However, the action space of the algorithm can only be discrete values, and it is not suitable for the continuous action space, which has significant limitations. Therefore, Xu et al. [15] proposed to use the Asynchronous Advantage Actor-Critic (A3C) to solve the computational task offloading model and effectively proved that the algorithm can have unique advantages and achieve convergence quickly. Based on this, this paper proposes the Deep Deterministic Strategy Gradient (DDPG) with the same dual network structure to solve the optimization problem of computational offloading in the perception layer network. According to the network changes of the perception layer, the computing task offloading strategy of the IoT terminal is adaptively adjusted, and the task delay is minimized based on ensuring the task completion rate. Finally, this paper compares the algorithm with DQN, A3C and other advanced algorithms from many aspects and proves the superiority of the algorithm.

The remaining paper is organized as follows. Section 2 mainly introduces related work, including the structure and offloading decision of Mobile Edge Computing (MEC). Section 3 proposes an IoT-oriented MEC offloading algorithm. Simulation experiments and results analysis are given in Section 4. Finally, Section 5 presents the conclusion.

## 2. Related work

### 2.1 MEC infrastructure

The composition of MEC usually includes three parts: IoT terminal layer, edge server layer and cloud data center layer [16]. As shown in Fig 1, the terminal layer of the IoTs has smart cars, smartphones, and various types of sensors with specific processing performance; the edge server layer is divided according to relative distance, and each area is divided Contains an edge server with moderate performance; unlike edge servers, the cloud data center layer contains many high-efficiency physical servers, which are gathered to form a cluster to provide services for users [17]. When the task from the IoTs terminal needs to be uninstalled, first divide the entire mobile application into subtasks that have data interaction with each other but can be executed independently by a particular sorting algorithm. Because when dealing with these tasks, some sub-tasks can only be executed locally. Others are tasks that can be offloaded, usually data processing tasks with a large amount of calculation. In this architecture, the networking scenarios where IoT devices are located: for example, wired and wireless network scenarios, are different; the characteristics of the tasks to be performed on the device (such as traffic characteristics and time delay characteristics) are also different; users pass Various IoT devices connect to the network in different ways. These devices generate tasks to which they are connected and send requests to edge subnets. The agent of the autonomous edge subnet will obtain the network status of other subnets through the distributed file system. After receiving the task request from the IoTs device, the agent makes calculation and offloading decisions based on the characteristics of the request. After training, computationally intensive tasks are often executed on cloud servers, while time-sensitive tasks are usually performed on local and edge servers.

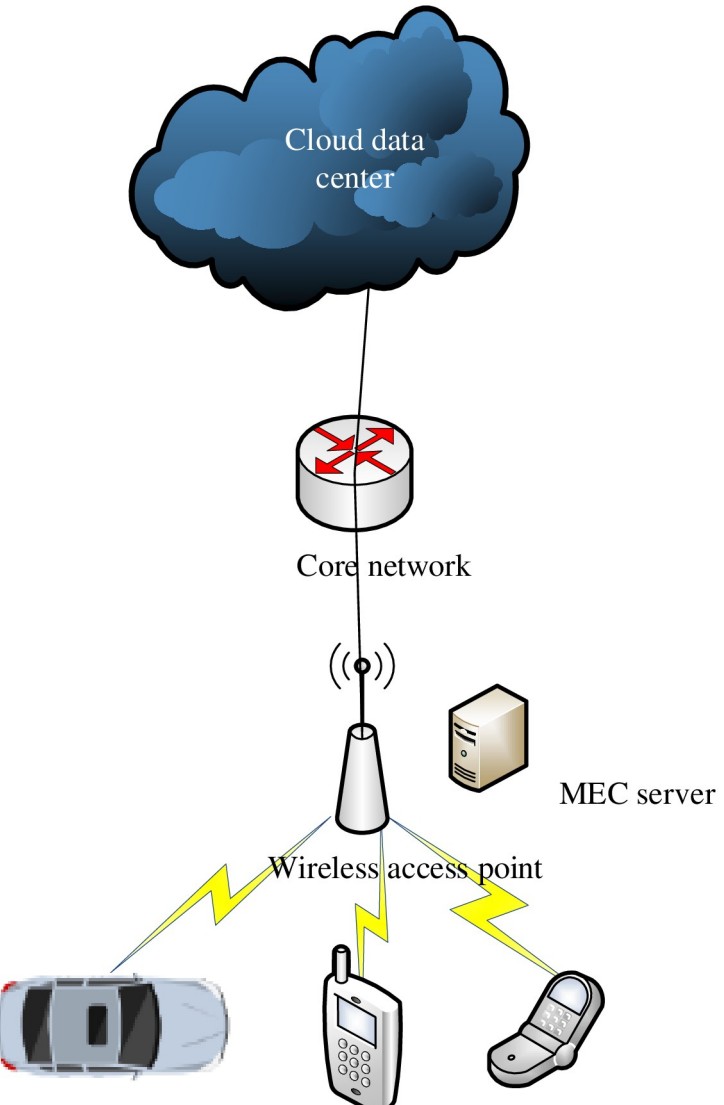

**Fig 1. MEC three-layer architecture.**

## 2.2 Offloading decisions in MEC

The uninstallation decision means that our user terminal decides whether to uninstall, how much to uninstall and what to uninstall. Generally, the results of the uninstallation decision made by the user terminal are divided into three situations, namely, local execution, full uninstallation and partial uninstallation [18–21]. As shown in the Fig 2.

1. Local execution: As shown in the smart Device 1 in Fig 2, when the overhead of offloading the task to the MEC server is too large, or the MEC server has no available resources. The entire task calculation process can only be executed locally;

2. Full uninstallation: As shown in the smart Device 2 in Fig 2, complete uninstallation means that all tasks are uninstalled to MEC for calculation;

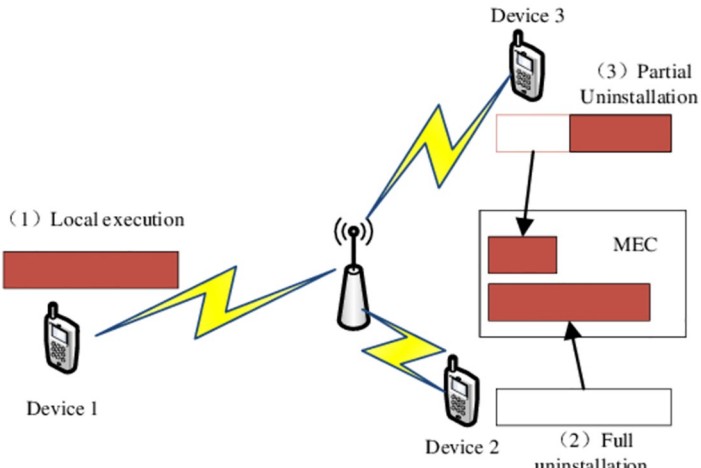

**Fig 2. Offloading decision.**

3. Partial uninstallation: As shown in the smart Device 3 in Fig 2, when the computing task can be split, part of the task can be calculated locally as needed, and the remaining part can be uninstalled to the MEC server for calculation.

## 3. IoT-oriented offload architecture and algorithms for edge computing

There are generally multiple heterogeneous wireless sensor networks in the perception layer of the IoT to meet the needs of different monitoring tasks. Each IoT device can join the appropriate wireless sensor network according to task attributes. Moreover, mobile IoT devices can choose a suitable wireless sensor network according to their location. Based on edge computing, a wireless sensor network can be regarded as an edge subnet in the IoT. The MEC in the edge subnet accepts the offloading request of the IoT terminal in the network, makes an offloading decision, and coordinates the calculation of the resources of the subnet and other subnets to maximize the computing resources of the perception layer network [22–27].

As shown in Fig 3, for the edge computing-oriented perception layer network, this paper proposes a computing offloading decision based on Deep Reinforcement Learning (DRL), which explicitly includes information input flow and decision output flow. The information input flow is the information flow from the edge network to the decision optimization engine based on DRL. The decision output flow is the decision control flow from the decision optimization engine to the edge network entity.

IoT devices access the edge network through different access methods and request tasks to be offloaded upwards. At this time, MEC extracts the feature information of the task, extracts an abstract security descriptor suitable for the task feature of the decision engine, and outputs it to the decision optimization engine [28–30]. When the IoT device generates a task and sends a service request to the network, the request composed of the feature vector of the task will be captured by the agent in the subnet. In the IoT-oriented edge computing offload architecture proposed in this paper, the coordination between edge servers is realized through a distributed file system. The agents of each edge subnet can obtain real-time network status through the distributed file system [31]. Therefore, the agent can make correct decisions for computing offloading. Agents in the edge network offload tasks to appropriate locations in the network for processing, thereby achieving overall performance optimization. After obtaining the network

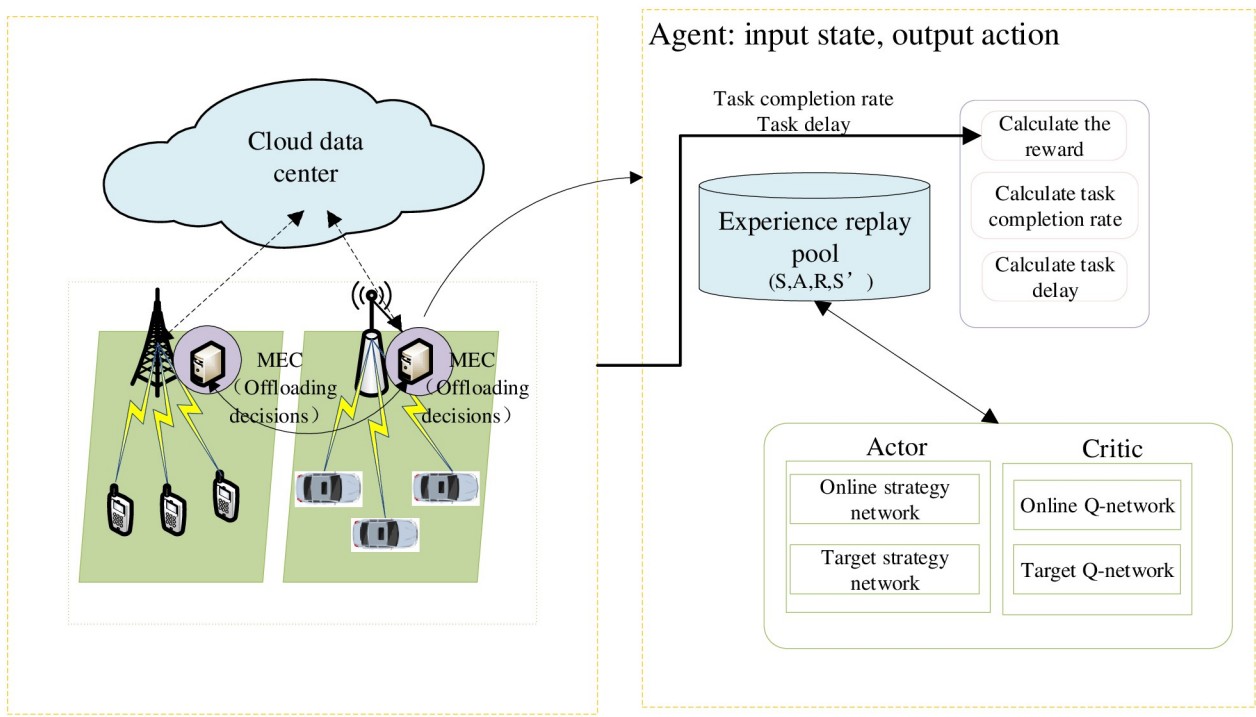

**Fig 3. Procedural flowchart of the offloading decision.**

state matrix and task feature vector, the decision-making optimization engine decides on computing offloading and deploys the physical instance through the edge network agent [32].

## 3.1 Formal descriptions

This paper aims to find a computation offloading strategy that expects minimum task delay while guaranteeing task load factor, where the task delay depends on the greater local execution and offloading of the computation. Table 1 gives a description of the parameters used in this paper.

**3.1.1 Characteristics of edge network.**   The edge network is composed of IoT devices $N_D$, and each device is connected to a different MEC according to its communication characteristics. It is an excellent choice to offload tasks to the edge server reasonably. In this process, each MEC will perform many tasks from IoT devices. We set the task waiting for the queue of the edge server, $q_{e_i}$, which indicates that there is a task $i$ with time $t$ that needs to be processed. Here, $e$ is the edge server. If $q_{e_i}$ of the MEC is very long, it means that the MEC is overloaded, and the task will be offloaded to the cloud server for processing [33, 34].

This framework assigns tasks to local, edge and cloud servers based on policies. For example, the Access Edge Computing Server (aECS) assigns tasks to the Neighboring Edge Computing Server (nECS) for collaborative processing based on the network state and task characteristics. When nECS completes the task, the task will be returned to aECS, and aECS will be integrated and returned to the IoT device.

The state of each level can be expressed as a state set $S(S^E, T^d, S^d, L^{E \to E})$, which is explained as follows. $S^E$ represents the state set of the edge server, $S^E = \{S_j = (q_{e_j}, f_{e_j}) | j = 1, 2, \ldots, m\}$. Here, $q_{e_j}$ is the task queue duration of the computing server $j$, $f_{e_j}$ is the frequency of the CPU in

**Table 1. A description of the parameters.**

| Parameters | Description |
|---|---|
| $S^E$ | The state set of the edge server |
| $T^d$ | The feature set of the task |
| $S^d$ | The status and characteristics of the IoT devices |
| $L^{E \to E}$ | The communication link bandwidth between nodes |
| $q_e$ | Task queue duration of MEC |
| $q_d$ | Task queue duration of IoT devices |
| $c_i$ | Calculation cycle |
| $t_i$ | Task completion time |
| $E$ | Expected delay |
| $n$ | Total number of task |
| $M$ | Total number of MECs |
| $M_i$ | set of available MECs for task $n_i$ |
| $M_{ij}$ | Task $n_i$ is executed on MEC $M_j$ |
| $B_{ij}$ | Task execution time of task $n_i$ on MEC $M_j$ |
| $R$ | Reward |
| $G$ | Total reward |
| $\pi_\theta(s)$ | Strategy function |
| $Q(s, a)$ | Value function |
| $\phi(S)$ | The current state vector |
| $\eta$ | Noise |
| $\theta$ | Actor network parameter |
| $\omega$ | Critic network parameter |

the edge computing server $j$, where $q_{e_j}$ is given as follows.

$$q_{e_j} = \sum_{i=1}^{N} \frac{c_i * a_i^{d \to e}}{f_{e_j}} \tag{1}$$

$T^d$ represents the feature set of the task, $T^d = \{c_i, t_{d_i}, s_i^u, s_i^d | i = 1, 2, \ldots, N\}$, $c_i$ represents the calculation period required to process task $i$, $t_{d_i}$ represents the deadline of task $i$, and $s_i^t$ is the size of task $i$.

$S^d$ represents the status and characteristics of the IoT devices, $S^d = \{S_j = (q_{d_j}, f_{d_j}) | j = 1, 2, \ldots, n\}$, $q_{d_j}$ is the task queue duration of the IoT device $d_j$, $f_{d_j}$ is the frequency of the edge computing server $d_j$ CPU, where $q_{d_j}$ is given as follows:

$$q_{d_j} = \sum_{i=1}^{N} \frac{c_i * a_i^d}{f_{d_j}} \tag{2}$$

$L^{E \to E}$ represents the communication link bandwidth between nodes.

**3.1.2 Offloading action.** The task offloading operation includes three parts: from the IoT device to aECS, from aECS to nECS, and from aECS to cloud server [35]. Unloaded tasks include task execution code and task data. It has been found that the unloading ratio of the computing cycle is roughly equal to the total amount of task unloading, so we merge the two sub-actions of IoT device unloading into aECS. Then a set of calculation actions in each episode can be expressed as $A(A^{d \to \hat{e}}, A^{\hat{e} \to e}, A^{\hat{e} \to \breve{e}})$, as follows:

$A^{d\to\hat{e}}$ represents the proportion of task calculation, or data volume offloaded from IoT devices to aECS.

$A^{\hat{e}\to e}$ represents the proportion of task calculation or data volume offloaded from aECS to cECS.

$A^{\hat{e}\to e}$ represents the proportion of task calculation or data volume offloaded from aECS to the cloud server.

### 3.1.3 Calculation model.

1. Local execution
   The local computing time consists of two parts: the local execution time and the local task queue waiting time. The total completion time of the local calculation task is calculated by the following formula:

$$t^{local} = a_i^d \times \frac{c_i}{f_{d_j}} + q_{d_j} \tag{3}$$

2. Service execution
   If all tasks are performed locally, the computing power of the IoT device is insufficient to complete the task within the deadline, so the IoT device needs to offload part of the task to the edge server [36]. Calculating the time spent on unloading includes task queuing time, task execution time, and task transmission time. For example, the following formula calculates edge calculation time + queuing time.

$$t_i^{cal+que} = \max(a_i^{d\to\hat{e}} \times \frac{c_i}{f\hat{e}} + q\hat{e}, a_i^{\hat{e}\to e_j} \times \frac{c_i}{f_{e_j}} + q_{e_j}, a_i^{\hat{e}\to e} \times \frac{c_i}{f_{e_j}} + q\hat{e} | j \in \{1, 2, \ldots, m-2\}) \tag{4}$$

The calculation formula of task transmission delay is as follows:

$$t_i^{trans} = \max(a_i^{d\to\hat{e}} \times \frac{S_t}{l^{d\to\hat{e}}}, a_i^{\hat{e}\to e_j} \times \frac{S_t}{l^{\hat{e}\to e_j}}, a_i^{\hat{e}\to e} \times \frac{S_t}{l^{\hat{e}\to e}} + q\hat{e} | j \in \{1, 2, \ldots, m-1\}) \tag{5}$$

3. Optimization goals
   Our optimization goal considers the expected task delay and the task completion rate (TCR). According to the above formula, the task completion time is:

$$t_i = \max(t_i^{local}, (t_i^{cal+que} + t_i^{trans})) \tag{6}$$

In addition, we need to consider some constraints when calculating. $n$ represents the total number of tasks, and $M$ represents the total number of MECs. $i$ represents the task index, and $j$ represents the MEC index. $M_i$ denotes the MEC-available set of task $n_i$, and $M_{ij}$ denotes that task $n_i$ is executed on MEC $M_j$. $t_i$ represents the task completion time of task $n_i$. $B_{ij}$ represents the task execution time of task $n_i$ on MEC $M_j$.

$$X_{ij} = \begin{cases} 1, & \text{if task } n_i \text{ is executed on MEC } M_j \\ 0, & \text{others} \end{cases} \tag{7}$$

$$\sum_{j \in M_j} X_{ij} = 1, \forall i, j \tag{8}$$

$$B_{ij} \leq t_i \tag{9}$$

$$B_{ij} \geq 0, t_i \geq 0 \tag{10}$$

Among them, constraint (8) indicates that one MEC can be selected from the set of optional MECs for each task $n_i$ for execution, but each task can only be executed on one MEC. Constraint (9) restricts that the task execution time of each task $n_i$ on MEC $M_j$ cannot exceed its task completion time. constraint (10) qualifies the non-negativity of all parameters. Assuming that there are P tasks in total, the expected delay of all tasks (which can also be understood as the average task completion time of P tasks) can be expressed as

$$E_t = \frac{\sum_{i=1}^{P} t_i}{P} \tag{11}$$

Assuming that there are C tasks completed within the deadline, the task completion rate of all tasks can be expressed as:

$$TCR = \frac{C}{P} \times 100\% \tag{12}$$

This paper aims to maximize the TCR while making the task waiting time as small as possible. Therefore, when the task is not completed within the deadline, a negative reward is given, and when the task is completed within the deadline, a high positive reward is given. We set the reward as:

$$R_i = \begin{cases} -t_i, & t_i > t_{d_i} \\ log_{0.995}(1 - \frac{1}{e^{\sqrt{t_i}}}), & t_i < t_{d_i} \end{cases} \tag{13}$$

Where $t_{d_i}$ represents the task deadline.

To explain the role of rewards more clearly, we assumed that the deadline of a task $n_i$ is 9ms, and the execution time of a task in one training is 10ms. Hence, the reward generated by this state is -10. In the next iterative training, the execution time of task $n_i$ is 8 ms, so the state reward is 12.15. After this iteration, the DRL agent finds that the total return of the second iteration is greater than the previous iteration, and the neural network will remember the actions in this iteration. In another iteration of training, the execution time of task $n_i$ is 6ms, and the reward for this state is 18.01, which is greater than the previous two iterations. Through continuous training, the overall computational load of the decision-making agent will perform better.

## 3.2 Offloading algorithms for IoT-oriented edge computing

In this section, we abstract the offloading of complex calculations as Markov Decision Processes (MDPs) [37, 38]. Reinforcement Learning (RL) is an algorithm for sequential decision-making. It continuously conducts trial and error learning in the target environment and continuously changes the strategy through the feedback of environmental information, seeking the greatest reward under this strategy. However, it has many advantages, yet is challenging to expand and suitable for relatively low-dimensional problems. Therefore, to solve this problem, DRL is considered. DRL combines the advantages of deep learning and RL to solve the problem of high-dimensional state space and action space [39–42]. In this paper, by abstracting the complex IoT computational offloading problem into a Markovian decision process and

combining multi-intelligent DRL algorithms, a collaborative environment based on IoT-oriented multi-intelligents is constructed to solve the edge server offloading problem for IoT terminals.

**3.2.1 Markov decision process.** The optimization goal is a single time slot object, which only depends on the current state [43]. However, the state of the network environment is dynamic, so the past network state is also an important reference factor for calculating offloading actions [44, 45]. If only the current state is used to make a decision, the decision-making behavior of the agent will lack further vision.

RL is a method of optimizing problems in a dynamic environment. We first formulate the problem as an MDPs, represented by a four-tuple $\{S, A, P, R\}$, where $S$ is the state space of the model, $A$ is the action space of the model, $P$ is the state transition probability of the model, and $R$ is the reward function of the model [46–48]. The description of each element is as follows:

1. State-space: The state space is defined as the state of each server and task in the edge network. The server set is $S = \{s_1, s_2, \ldots s_n\}$, where $n$ is the number of servers in the edge network. The server status includes the length of the task waiting for a queue. The task set is $A$, where $v$ is the total number of tasks. Task status $S_M = \{m_1, m_2, \ldots m_v\}$, $q_i \in [0, 1]$, $\sum_{i=1}^{n} q_i = 1$ represents the percentage of tasks allocated by the server at each location.

2. Action space: At each moment, after considering the length of the task waiting for the queue of each node and the deadline of the task itself, the agent must make an action to allocate the task to each server for processing. We define the action space as

$$A(A^{d \to \hat{e}}, A^{\hat{e} \to e}, A^{\hat{e} \to e}) \tag{14}$$

The constraints are as follows.

$$A^{d \to \hat{e}} + A^{\hat{e} \to e} + A^{\hat{e} \to e} = 1 \tag{15}$$

The meanings of the letters have been explained in the previous section.

3. Reward function: Whenever an agent makes an action, the environment will automatically give a reward: here we define the reward value as formula (8), hence, the total reward is $G = \sum_{t=1}^{T} R(s'_t, a'_t, s_{t+1})$, and our final goal is to maximize the total reward.

In addition to the above four elements, a hyperparameter $\gamma$, $\gamma$ is the future reward weight, and the value range is [0, 1]. When $\gamma$ tends to 0, the value function focuses on the current reward. And if $\gamma$ tends to 1, the value function will consider more rewards from subsequent steps. In other words, $\gamma$ makes decisions that favor short-term or long-term returns.

**3.2.2 Dynamic resource optimization algorithm based on DRL.** In resource allocation decision-making, we need to interact with the environment to obtain samples directly. The ultimate goal of the sample estimated value function is to find the optimal strategy $\pi^*$. The network model's state and action space proposed in this paper are high-dimensional, dynamic and non-discrete. Therefore, they need to be optimized in the sequence generation process. We choose the DDPG method based on the above factors to optimize our decision-making algorithm. DDPG is derived from an improved version of the actor-critic and strategy gradient algorithm and draws on the dual network structure of Double Deep Q Network (DDQN) [49].

As shown in Fig 3, the deployment of DRL consists of two parts: the network environment and the agent. The network environment comprises network nodes (IoT nodes and cloud nodes), network monitors and users. The network node receives the user's task offloading request. The network monitor collects the information in the network in real-time and interacts with the agent information to respond to the state changes in the network.

DDPG uses a strategy function $\pi_\theta(s)$ to make decisions. It explicitly maps a state to a specific action. This can greatly improve the convergence of training. In the actor-critic model, the agent uses the strategy gradient method to enhance the gradient and selects the operation in the current state with the highest probability through the strategy function $\pi_\theta(s)$. Correspondingly, the critic network evaluates the current decision based on the time error between the value function and the current reward and evaluates the actor's behavior. The formula for calculating the deterministic policy gradient is:

$$\nabla_\theta J(\pi_\theta) = E_{s \sim \rho^\pi}[\nabla_\theta \pi_\theta(s) \nabla_a Q_\pi(s, a) | a = \pi_\theta(s)] \tag{16}$$

As shown in Fig 3, the distributed dynamic resource optimization algorithm based on DRL consists of four parts: edge network environment, experience replay pool, dual actor-network, and dual critic network, of which two networks of actors and two networks of critics have the same structure. The network control node interacts with the edge network environment to obtain the current network state and stores the current state vector $\phi(S)$, action vector $A$, reward $R$ and the next state vector $\phi(S')$ of the experience pool. To explore the action space more extensively, we add some noise to the actions chosen by the actors. Then, after accumulating a certain amount of data in the experience pool, take out the small batch size data block and input it into the estimated neural network to obtain the action-value function. The loss function is calculated together with the action-value function of the target value network, and the gradient is reversed. Transmit to update all the parameters $\omega$ of the current network, where the loss value function can be expressed as:

$$\frac{1}{m} \sum_{j=1}^{m} (y_j - Q(\phi(S_j), a_j, \omega))^2 \tag{17}$$

We define the current target $Q$ value, which is used to calculate the expected reward value of the action in the current state. We define how to update the actor-critic neural network parameters:

$$y_j = R_j + \gamma Q_{target}(\phi(s'_j), \pi_{\theta'}(\pi(s'_j)), \omega') \tag{18}$$

This represents the weighted expectations of the current reward status and possible future rewards and is used to evaluate the value of the current status. We define how to update the actor-critic neural network parameters:

$$\begin{cases} \omega' \leftarrow \tau\omega + (1 - \tau)\omega' \\ \theta' \leftarrow \tau\theta + (1 - \tau)]\theta' \end{cases} \tag{19}$$

This algorithm does not directly copy the parameters of the target network to the evaluation network. Still, it uses a gradual update method, and only a small amount of each parameter is updated. At the same time, to increase the randomness of the learning process and better explore the entire solution space, we added some noise to the learning process. The action selection expression is defined as follows:

$$A = \pi_\theta(S) + \eta \tag{20}$$

where $\eta$ is noise. Next, we define the loss functions of the critic and actor networks. The loss

function of the critic network is defined as follows.

$$J(\omega) = \frac{1}{m} \sum_{j=1}^{m} (y_j - Q(\phi(S_j), a_j, \omega))^2 \tag{21}$$

Refer to (21) for the loss function of the actor-network. The defined loss gradient is as follows.

$$\nabla_\theta J(\pi_\theta) = E_{s \sim \rho^\pi}[\nabla_a Q_\phi(s,a)|_{s=s_i, a=\pi_\theta(s)} \nabla_\theta \pi_\theta|_{s=s_i}] \tag{22}$$

The computational offloading algorithm based on DDPG is as described in Algorithm 1. It consists of two parts: the initialization of the network environment and the DRL algorithm in the agent. Initially, the decision made by the agent is close to a random algorithm. However, with the learning process of the computational offloading algorithm based on DDPG, the resulting offloading strategy is getting closer and closer to the optimal algorithm. After the learning iteration is over, the learned DDPG neural network parameters are obtained.

From the pseudo-code of Algorithm 1, it can be seen that the time complexity of this algorithm is related to the size of the episode and the size of the time step in each episode. Line 2 takes an average of $t_{11}$ time to run once, and runs $N$ times repeatedly, so the running time is $t_{11}N$. Assume that running 4 to 13 rows in each time step takes an average of $t_{12}$ time. Running $T$ times consumes an average of $t_{12}T$ time and then repeats the operation $N$ times. So the running time is $t_{12}TN$. So the total time to run is $t_{11}N + t_{12}TN$. When the values of $N$ and $T$ are relatively large, the coefficients of $t_{11}N$ and $t_{12}TN$ are negligible, so the time complexity is $O(N + NT)$. Because $NT$ grows much faster than $N$, the time complexity of this algorithm is $O(NT)$.

**Algorithm 1** Edge Offloading Computing Based on DDPG

```
Input: environmental parameters;
Initialization: The parameters of the actor online strategy network
and the target strategy network are respectively ω, ω′; the parameters
of the actor online strategy network and the target strategy network
are respectively and θ, θ′;
1: for episode = 1, N do
2:   Initialize the edge state and task queue
3:   for t = 1, T do
4:      Obtain the current state S_t from the edge network environment
and convert it into a vector φ(S_t)
5:      Unload tasks according to the strategy π_θ(φ(s)) in the actor-
network
6:      Calculate the reward according to the formula (13)
7:      Get the next state s_(t+1)
8:      Store (φ(s_t), a_t, r_t, φ(s_{t+1})) in the experience replay pool
9:      Calculate the Q value of y_i by formula (18)
10:   Use the formula (22) to update the actor strategy
11:   Use the formula (19) to update the actor target network
12:   Use the formula (19) to update the critics' target network
13:   Let t = t + 1
14:   end for
15: end for
Output: Current optimal actor network parameter θ, critic network
parameter ω
```

# 4 Simulation experiments

## 4.1 Heterogeneous perception layer network

In this section, the DDPG-based uninstall strategy algorithm will be evaluated in the MEC task uninstall scenario of multiple edge servers, scenarios, and IoT terminals. The experiments in

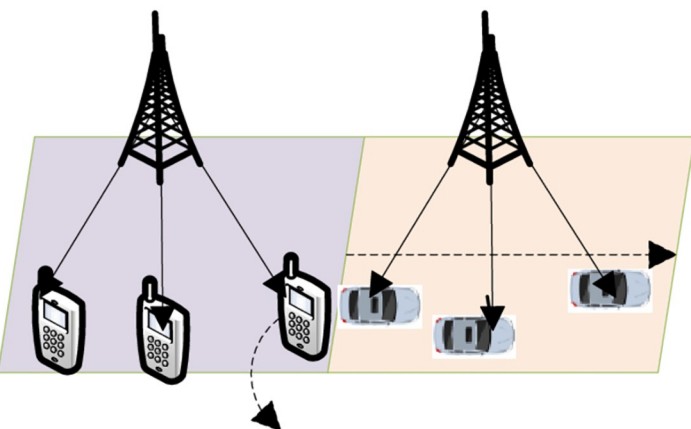

**Fig 4. Simulation of different access scenarios.**

this paper are all implemented based on Python 3.8 programming under the Windows 10 system, and the DDPG algorithm is implemented based on the TensorFlow framework programming. The MEC server is equipped with an Intel Core i7-7700 CPU. In the standard configuration, 30 IoT devices are connected in each case. In the wireless access scenario, as shown in Fig 4, we divide the scenarios into two categories: a wireless smart terminal and a vehicle-mounted IoT device. Pedestrians are driving in a low-speed environment, and vehicles are driving at a constant speed (0-50m/s) in a fixed direction at high-speed [50].

## 4.2 Simulation settings

The simulated environment is a three-layer edge network, including the IoT terminal layer, edge server layer, and cloud data center layer described above. There are 2-8 wireless edge servers at the edge server layer. The service range of each wireless AP is 50×50m, and the speed of each wireless mobile IoT device is randomly 0-40m/s. Initially, there are IoT devices within the service range of each AP, and the probability of each IoT device generating unloading requests in batches at each time satisfies the Poisson distribution [51, 11].

Both the actor-network and the critic network of DDPG have a three-layer structure, and the second layer of the fully connected layer has 200 neurons. The input state vector is normalized in the first layer. The relevant parameters in the experiment are shown in Table 2.

## 4.3 Algorithm comparison

Four settings were changed in this experiment to study the factors affecting the algorithm's performance. In addition, four other offloading algorithms are compared with the proposed algorithm in each scenario, as follows:

1. DQN-based offloading algorithm [14]
   Since the action space of DQN can only be discrete values, we set the action in each dimension of DQN to 0.2. The IoT device requests the execution of computational offloading for each task according to the decisions given by the well-trained DQN network. Moreover, to change the number of columns, select the Columns icon from the MS Word Standard toolbar and select the correct number of columns from the selection palette.

**Table 2. Parameters setting of simulation experiment.**

| Parameters | Description | Value |
|---|---|---|
| $s^t$ | Data size of task | 100KB-2MB |
| $t_d$ | Deadline of task | 5ms-50ms |
| $c$ | CPU calculation cycle of the task | 0.1 |
| $f_d$ | CPU frequency of IoT devices | 0.5GHZ |
| $f_e$ | CPU frequency of MEC | 3-5GHZ |
| M | Large amounts of MEC | 3-12 |
| $N_D$ | Number of IoT devices | 6 |
| $L^{d \rightarrow e}$ | Bandwidth from IoT device | 50MB/s |
| $L^{E \rightarrow e}$ | Bandwidth between MECs | 300MB/s |
| $L^{e \rightarrow \bar{e}}$ | Bandwidth from MEC to cloud server | 200MB/s |
| episode | Number of iterations | 6000 |
| aA | Actor network learning rate | $1 \times 10^{-3}$ |
| aC | Critic network learning rate | $2 \times 10^{-3}$ |
| Mem | Experience pool size | 10000 |
| Batch | Small batch size | 64 |
| $\gamma$ | Reward discount rate | 0.9 |

2. A3C-based offloading algorithm [15]
   Both A3C and DDPG are actor-critic improved algorithms. It has unique advantages, such as the convergence of algorithm training. This algorithm has recently been applied in many studies, which is why we choose this algorithm as an experimental comparison algorithm.

3. Edge server computing
   The IoT terminal ignores the burden of the edge server and always offloads tasks to the edge server for calculation.

4. Local computing
   The IoT terminal puts all computing work locally and does not request that the computing be offloaded to the edge network.

## 4.4 Simulation result

In this section, we compare the performance of this algorithm with other algorithms in heterogeneous network scenarios. As shown in Figs 5 and 6, the proposed algorithm is compared with other algorithms in different settings of the edge network environment. The CPU capacity is changed in Fig 5, and the number of MECs is changed in Fig 6. We can see that the performance of the proposed algorithm and the A3C-based algorithm is significantly better than the other algorithms. Since both DDPG and A3C algorithms belong to the actor-critic model, they perform very close in each scenario. We set the capacity of each edge server to 2GHZ, 3GHZ, 4GHZ, 5GHZ, and 6GHZ. As the number of edge servers increases from 3 to 12, the average task delay of the computation methods other than local computation gradually decreases. After the number of servers is increased to 6, increasing the number of edge servers does not significantly improve the results. It can be seen that the average task delay decreases as the CPU capacity of the edge servers increases. But this improvement is not significant compared to the increase in CPU capacity. The task delay is limited by the network bandwidth and the endpoint's CPU capacity, and there is a theoretical limit. In addition, $s^t$ represents the data

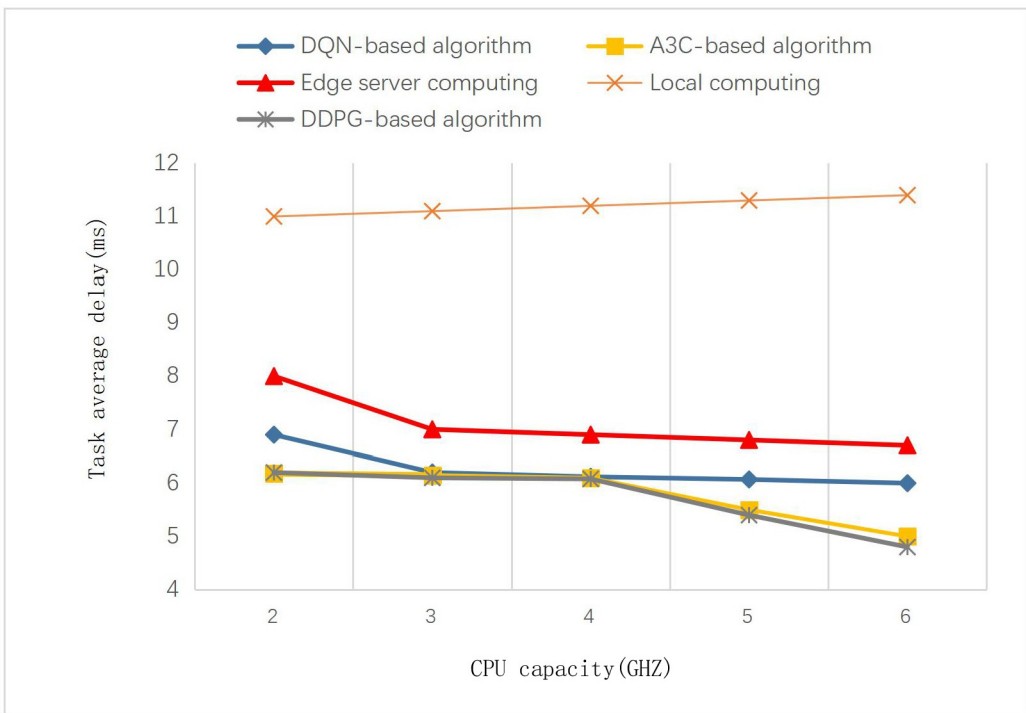

**Fig 5. Impact of edge servers with different CPU capacity on task average delay.**

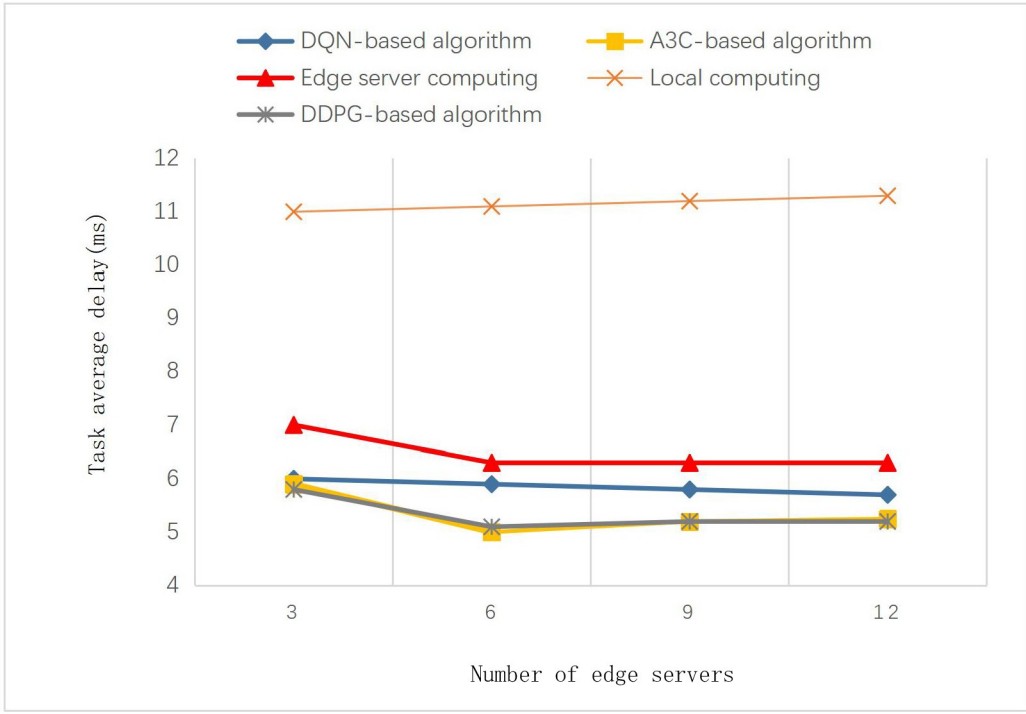

**Fig 6. Impact of different number of edge servers on average task delay.**

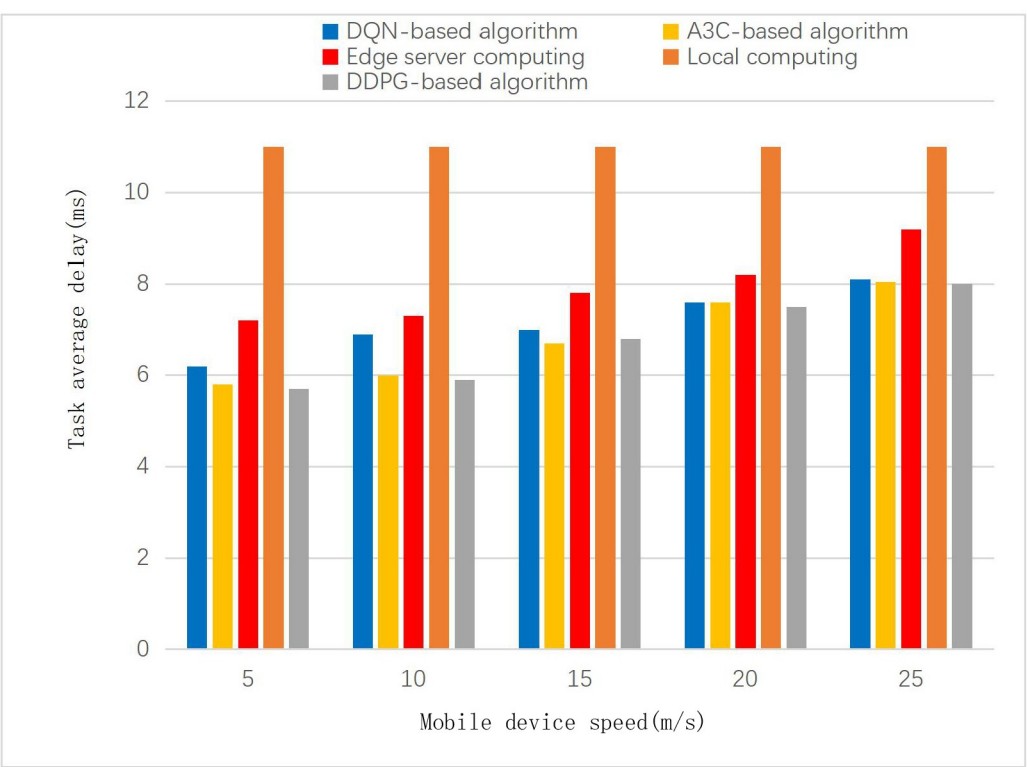

**Fig 7. Impact of mobile IoT devices at different speeds on average task delay.**

size of the offloaded task. The set data size is within the acceptable calculation range of the mobile edge server. When the task is offloaded, it can adaptively find a suitable edge server for calculation. It won't affect the average delay too much.

In the MEC scenario, IoT terminals move from the service area of one access point to the service area of another. Due to migration, the speed of movement of mobile IoT devices also affects the performance in this scenario. According to Fig 7, in addition to the local execution of IoT devices, task wait times are expected to increase as IoT devices move faster. Under low-speed conditions, the performance of the DDPG-based algorithm is significantly better than the other offloading modes. As the speed increases to 40m/s, the performance of the DDPG-based algorithm is only slightly better than that of DQN and A3C.

The impact on the total IoT terminal service delay expectation is shown in Fig 8. It can be seen that the DDPG-based algorithm performs optimally and increases the task delay expectation as the number of IoT terminals increases. In the mobile edge computing scenario, the increase in the number of IoT terminals leads to an increase in the number of task requests in the time slot, which increases the burden on the edge server.

The experiments in this paper mainly compare the edge network environments with different settings. As a result, it is verified that the DDPG algorithm can effectively reduce the task delay from multiple perspectives, such as different computing capabilities of edge servers, different numbers of edge servers, IoT devices at different speeds, and different numbers of IoT devices. Furthermore, it has been proved that the algorithm can effectively solve the optimization offloading of computing tasks of IoT terminals compared with other algorithms.

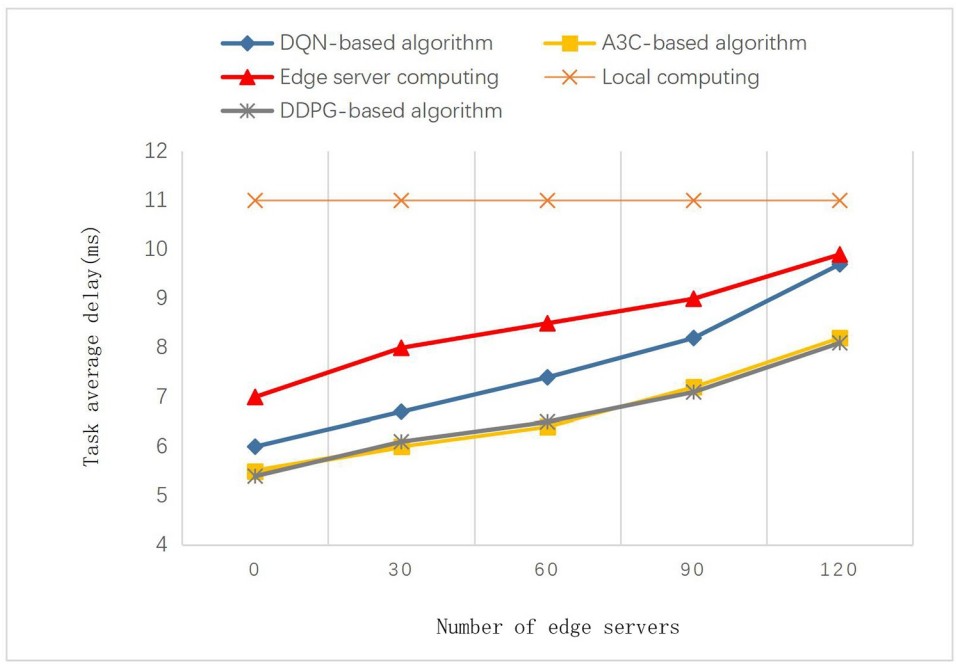

**Fig 8. Impact of different number of IoT devices on average task delay.**

## 5 Conclusions

The extensive distribution mode and powerful offloading capability of edge computing nodes greatly facilitate the task computation and data transmission of IoT terminals. In this paper, the DDPG algorithm is proposed to solve the computational offloading problem of IoT terminals for perception layer networks. Experiments show that the DDPG algorithm proposed in this paper can effectively improve the operational efficiency of perception layer networks and reduce the average delay of tasks.

## Author Contributions

**Conceptualization:** Yongli Xing, Muhammad Waqas, Hisham Alasmary.

**Data curation:** Yongli Xing.

**Formal analysis:** Yongli Xing, Sami Ullah, Muhammad Waqas.

**Investigation:** Yongli Xing, Muhammad Waqas.

**Methodology:** Yongli Xing, Sami Ullah, Muhammad Waqas.

**Project administration:** Muhammad Waqas.

**Resources:** Tao Ye, Sami Ullah, Muhammad Waqas, Hisham Alasmary.

**Supervision:** Tao Ye, Muhammad Waqas, Zihui Liu.

**Validation:** Tao Ye, Sami Ullah, Muhammad Waqas, Hisham Alasmary, Zihui Liu.

**Visualization:** Tao Ye, Sami Ullah, Muhammad Waqas, Zihui Liu.

**Writing – original draft:** Yongli Xing, Sami Ullah.

**Writing – review & editing:** Tao Ye, Sami Ullah, Muhammad Waqas, Hisham Alasmary, Zihui Liu.

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
