## [Decision Letter · Decision Letter 0]

24 Jun 2022

PONE-D-22-12942A computational offloading optimization scheme based on deep reinforcement learning in perceptual networkPLOS ONE

Dear Dr. Waqas,

Thank you for submitting your manuscript to PLOS ONE. After careful consideration, we feel that it has merit but does not fully meet PLOS ONE’s publication criteria as it currently stands. Therefore, we invite you to submit a revised version of the manuscript that addresses the points raised during the review process.

We look forward to receiving your revised manuscript.

Kind regards,

Mahdi Abbasi, PhD.

Academic Editor

PLOS ONE

Journal Requirements:

 [NO - The funders had no role in study design, data collection and analysis, decision to publish, or preparation of the manuscript.]

5.  We noticed you have some occurrence of overlapping text with the following previous publication(s), which needs to be addressed:

- https://ieeexplore.ieee.org/document/9084152

[OPTIONAL: The text that needs to be addressed involves [INCLUDE BRIEF DESCRIPTION (e.g., the first two sentences of the Discussion.)]

In your revision ensure you cite all your sources (including your own works), and quote or rephrase any duplicated text outside the methods section. Further consideration is dependent on these concerns being addressed.

Reviewers' comments:

Reviewer's Responses to Questions

**Comments to the Author**

1. Is the manuscript technically sound, and do the data support the conclusions?

Reviewer #1: Yes

Reviewer #2: Partly

2. Has the statistical analysis been performed appropriately and rigorously? 

Reviewer #1: Yes

Reviewer #2: Yes

3. Have the authors made all data underlying the findings in their manuscript fully available?

Reviewer #1: Yes

Reviewer #2: No

4. Is the manuscript presented in an intelligible fashion and written in standard English?

Reviewer #1: Yes

Reviewer #2: Yes

5. Review Comments to the Author

Reviewer #1: This paper presents a nice idea on using deep reinforcement learning to investigate a computation offloading optimization scheme for the perception layer. I think it is an interesting approach, have enough novelty, and the paper is on the whole quite well written.

The issues to be addressed are:

Please clarify the difference between your proposed algorithm and the idea of reference [10].

Show the impact of data block size and block interval (from parameter settings) on average delay.

In addition to the average delay, show the impact of the parameters (at least one of them) on the throughput.

Reviewer #2: This paper investigates a computation offloading optimization scheme for the perception layer. The algorithm can adaptively adjust the computational task offloading policy of IoT terminals according to the network changes in the perception layer.

Although the work has potential, several major concerns need to be addressed before the paper can be accepted for publication:

- The optimization problem in Section 3.1.3 is not clear. Specifically, it is not clear from Equation (6) what the authors are trying to optimize. How can the task completion rate be optimized by maximizing the local and service execution times?

- Moreover, no constraints are defined for the optimization problem, which makes it incomplete.

- It is not clear what novel contributions the authors bring by employing Deep Reinforcement Learning (DRL) for offloading. Is the internal structure of DRL tuned for the considered problem or is it just used as a blackbox approach?

- Important literature on offloading/scheduling is missing such as:

- Ad hoc vehicular fog enabling cooperative low-latency intrusion detection. IEEE Internet of Things Journal, 8(2),

829-843 (2020).

- Intelligent task prediction and computation offloading based on mobile-edge cloud computing. Future Generation

Computer Systems, 102, 925-931 (2020).

- Multi-user multi-task computation offloading in green mobile edge cloud computing. IEEE Transactions on

Services Computing, 12(5), 726-738 (2018).

- A trust and energy-aware double deep reinforcement learning scheduling strategy for federated learning on IoT

devices. In International Conference on Service-Oriented Computing (pp. 319-333) (2020).

- What is the computational complexity of Algorithm 1? Can it be used to make real-time offloading decisions?

- The environment within which the simulations have been done needs to be further clarified, i.e., programming language, framework, resource specifications, etc.

6. PLOS authors have the option to publish the peer review history of their article (what does this mean?). If published, this will include your full peer review and any attached files.

Reviewer #1: No

Reviewer #2: No

---

## [Author Response · Author response to Decision Letter 0]

11 Aug 2022

Please find the attached pdf file.

---

## [Decision Letter · Decision Letter 1]

2 Jan 2023

A computational offloading optimization scheme based on deep reinforcement learning in perceptual network

PONE-D-22-12942R1

Dear Dr. Waqas,

We’re pleased to inform you that your manuscript has been judged scientifically suitable for publication and will be formally accepted for publication once it meets all outstanding technical requirements.

Kind regards,

Mahdi Abbasi, PhD.

Academic Editor

PLOS ONE

Reviewers' comments:

Reviewer's Responses to Questions

**Comments to the Author**

1. If the authors have adequately addressed your comments raised in a previous round of review and you feel that this manuscript is now acceptable for publication, you may indicate that here to bypass the “Comments to the Author” section, enter your conflict of interest statement in the “Confidential to Editor” section, and submit your "Accept" recommendation.

Reviewer #1: All comments have been addressed

Reviewer #3: All comments have been addressed

2. Is the manuscript technically sound, and do the data support the conclusions?

Reviewer #1: Partly

Reviewer #3: Yes

3. Has the statistical analysis been performed appropriately and rigorously? 

Reviewer #1: I Don't Know

Reviewer #3: Yes

4. Have the authors made all data underlying the findings in their manuscript fully available?

Reviewer #1: Yes

Reviewer #3: Yes

5. Is the manuscript presented in an intelligible fashion and written in standard English?

Reviewer #1: Yes

Reviewer #3: Yes

6. Review Comments to the Author

Reviewer #1: The authors have adequately addressed the comments and I feel that this manuscript is now acceptable for publication.

Reviewer #3: (No Response)

7. PLOS authors have the option to publish the peer review history of their article (what does this mean?). If published, this will include your full peer review and any attached files.

Reviewer #1: No

Reviewer #3: No

---

## [Editor Report · Acceptance letter]

15 Feb 2023

PONE-D-22-12942R1 

A computational offloading optimization scheme based on deep reinforcement learning in perceptual network 

Dear Dr. Waqas:

I'm pleased to inform you that your manuscript has been deemed suitable for publication in PLOS ONE. Congratulations! Your manuscript is now with our production department. 

Kind regards, 

on behalf of

Dr. Mahdi Abbasi 

Academic Editor

PLOS ONE